# Monitoring Treatment Response, Early Recurrence, and Survival in Uterine Serous Carcinoma and Carcinosarcoma Patients Using Personalized Circulating Tumor DNA Biomarkers

**DOI:** 10.3390/ijms24108873

**Published:** 2023-05-17

**Authors:** Stefania Bellone, Blair McNamara, Levent Mutlu, Cem Demirkiran, Tobias Max Philipp Hartwich, Justin Harold, Yang Yang-Hartwich, Eric R. Siegel, Alessandro D. Santin

**Affiliations:** 1Department of Obstetrics, Gynecology, and Reproductive Sciences, Yale University School of Medicine, New Haven, CT 06520, USA; stefania.bellone@yale.edu (S.B.); levent.mutlu@yale.edu (L.M.);; 2Department of Biostatistics, University of Arkansas for Medical Science, Little Rock, AR 72205, USA; siegelericr@uams.edu

**Keywords:** ctDNA, liquid biopsy, uterine serous carcinoma, uterine carcinosarcoma, biomarkers, early detection

## Abstract

Uterine serous carcinoma (USC) and carcinosarcomas (CSs) are rare, highly aggressive variants of endometrial cancer. No reliable tumor biomarkers are currently available to guide response to treatment or detection of early recurrence in USC/CS patients. Circulating tumor DNA (ctDNA) identified using ultrasensitive technology such as droplet digital polymerase chain reaction (ddPCR) may represent a novel platform for the identification of occult disease. We explored the use of personalized ctDNA markers for monitoring USC and CS patients. Tumor and plasma samples from USC/CS patients were collected at the time of surgery and/or during the treatment course for assessment of tumor-specific somatic structural variants (SSVs) by a clinical-grade next-generation sequencing (NGS) platform (i.e., Foundation Medicine) and a droplet digital PCR instrument (Raindance, ddPCR). The level of ctDNA was quantified by droplet digital PCR in plasma samples and correlated to clinical findings, including CA-125 serum and/or computed tomography (CT) scanning results. The genomic-profiling-based assay identified mutated “driver” target genes for ctDNA analysis in all USC/CS patients. In multiple patients, longitudinal ctDNA testing was able to detect the presence of cancer cells before the recurrent tumor was clinically detectable by either CA-125 or CT scanning. Persistent undetectable levels of ctDNA following initial treatment were associated with prolonged progression-free and overall survival. In a USC patient, CA-125 and TP53 mutations but not PIK3CA mutations become undetectable in the plasma at the time of recurrence, suggesting that more than one customized probe should be used for monitoring ctDNA. Longitudinal ctDNA testing using tumor-informed assays may identify the presence of residual tumors, predict responses to treatment, and identify early recurrences in USC/CS patients. Recognition of disease persistence and/or recurrence through ctDNA surveillance may allow earlier treatment of recurrent disease and has the potential to change clinical practice in the management of USC and CS patients. CtDNA validation studies in USC/CS patients prospectively enrolled in treatment trials are warranted.

## 1. Introduction

The rate of endometrial cancer is increasing among women living in industrialized countries; 66,200 newly diagnosed cases and 13,030 deaths are projected in the United States for 2023 [1]. Uterine serous carcinoma (USC) is a rare variant of endometrial cancer, accounting for 3% to 10% of all uterine carcinomas but responsible for 39% of all endometrial-cancer-associated deaths [2]. Mixed malignant Mullerian tumors (MMMTs) from the uterus, also known as carcinosarcomas (CSs), are biphasic tumors composed of epithelial and sarcomatous components [3]. Molecular evidence from our research group has recently unequivocally demonstrated that CSs originate as epithelial tumors and undergo mesenchymal transformation during tumor evolution [4]. While CSs account for only 3–5% of uterine cancers, similar to USC, they hold a disproportionately high mortality rate [3]. Standard treatment for USC and CS patients is based on aggressive surgical debulking followed by chemotherapy with or without radiation. Despite surgical cytoreduction and adjuvant therapy, 5-year survival rates remain poor for both these malignancies [2,3]. Novel targeted therapies and more sensitive and specific biomarkers for monitoring response to adjuvant treatment and for early detection of recurrent disease are desperately needed to improve USC and CS patient outcomes.

Multiple groups, including our own, have recently used whole exome sequencing (WES) to analyze the genetic landscapes of biologically aggressive gynecologic cancers, including USC and CSs [4,5]. Our reports, combined with the comprehensive genetic studies performed by The Cancer Genome Atlas (TCGA) network, have provided insights into the most common genetic alterations of these rare, highly lethal gynecologic tumors, including single genetic variants (SNVs) and copy number variations (CNVs). Importantly, advancements in sequencing technologies, combined with the development of ultrasensitive analytic techniques such as digital PCR, allow the detection and quantitation of minuscule amounts of cell-free DNA (cfDNA) in patient serum/plasma as a marker of disease (i.e., liquid biopsy). This cfDNA detection and quantification not only has the potential to provide an accurate reflection of minimal disease presence after the completion of surgical debulking and/or chemotherapy treatment but also represents a noninvasive strategy to evaluate tumor evolution and early detection of recurrence. This technology stands to benefit all cancer patients but has the potential to provide unique benefits in rare, biologically aggressive tumors such as USC and CSs, in which validated biomarkers are currently either not available or often unreliable.

In this retrospective study, we report on our experience using a rapid and efficient pipeline that combines the identification of tumor-specific point mutations in USC and CS patients using a clinically approved next-generation sequencing (NGS) assay (i.e., Foundation Medicine) [6] followed by the quantitation of targeted mutations using droplet digital PCR-based ctDNA to monitor tumor status during follow-up and treatment. The ctDNA results were compared against CT scans, CA-125 serum levels, and the known surgical/clinical status of patients. We demonstrate that ctDNA levels correlated well with clinical and radiologic outcomes in USC/CS patients and, in a few instances, were able to detect the presence of cancer cells earlier when compared with other currently recommended testing modalities. Prospective analyses to assess the utility of ctDNA as an early biomarker of clinical outcomes and early recurrence in USC and CS patients are warranted.

## 2. Results

### 2.1. Clinical, Pathologic, and Genetic Characteristics of Patients/Tumors

USC/CS patients undergoing routine treatment for gynecologic cancer were evaluated in this study, and their clinicopathologic information is shown in Table 1. Tumor-derived DNA and ctDNA and clinical data were available for all patients. Fourteen patients had USC histology, while 2 harbored a uterine CS (Table 1). Tumor biopsies were collected either at the time of primary surgery or at the time of recurrence and sequenced at Foundation Medicine as previously described [6].

### 2.2. Identification of SNV and Development of Personalized ddPCR Assays

Using the genetic results obtained using the FM platform (Table 1), we first aimed to identify potential tumor-specific “drivers” mutations for developing droplet digital PCR (ddPCR)-based assays. Of the mutations discovered using the targeted panel, the most commonly mutated gene in USC was TP53 in 15 out of 16 patients (Table 1). Mutations were also identified, albeit at lower frequencies, in PTEN, PIK3CA, PPP2R1A, KRAS, and FBXW7. As depicted in Table 1, multiple candidate mutations were identified for several of the patients, and ctDNA was extracted from 106 individual time points with an average (range) of 6.62, with 2–12 time point collections per patient (Table 1). In total, 18 unique assays were validated using quantitative PCR (qPCR). Next, we evaluated the ability of serially collected ctDNA to screen for the clinical presence of tumor and, whenever available, compared the results with CA-125, a test approved by the FDA for the detection of ovarian cancer but also commonly used for monitoring USC patients as well as CT imaging. Data comparing ctDNA and CA-125 levels to the presence of tumor on CT imaging were available for the majority of patients. Of interest, a lack of concordance between tests was observed in some of the USC patients. Importantly, as representatively described in the case reports below, all patients with detectable levels of ctDNA, regardless of the presence of positive or negative CA-125 results, were later found to have proven recurrent tumors that, in some cases, were initially undetected by CT scanning (Figure 1, Figure 2, Figure 3 and Figure 4).

### 2.3. Detailed Description of Representative Clinical Cases

Out of the 16 USC/CS patients studied with results presented in Table 1, we selected a few representative cases to describe in more detail the patients’ disease courses in relation to treatment response, biomarkers collection (i.e., ctDNA and CA-125), and CT imaging. Visual representations of the patients’ clinical courses with CA-125 and ctDNA trends can be found in Figure 1, Figure 2, Figure 3, Figure 4 and Figure 5.

Patient #1:

In October 2014, a 67-year-old female presented with postmenopausal bleeding and underwent robot-assisted total laparoscopic hysterectomy and bilateral salpingo-oophorectomy for a stage IA uterine serous carcinoma; notably, no lymph node or omental biopsies were taken at the time of staging. As depicted in Figure 1, the patient was then treated with adjuvant paclitaxel and carboplatin for six cycles, completed in March 2015, and received vaginal apex brachytherapy (three fractions, completed 30 May 2015). A CT scan performed in June 2015 showed no evidence of disease (NED). Six months later, a surveillance CT scan (18 December 2015) showed potential metastatic disease in the pelvis and vagina, which was also detected on PET CT (30 December 2015), and recurrent USC was confirmed on lymph node biopsy. At that time, as shown in Figure 1, CA-125 was normal at 12.7 U/mL (normal range: 0 to 35 U/mL), while ctDNA copy number was elevated to 770 mutant copy/mL. She was then enrolled in a clinical trial (NCT01367002), “Randomized Phase II Evaluation of Carboplatin/Paclitaxel with and without Trastuzumab in HER2/neu+ Patients with Advanced/Recurrent Uterine Serous Papillary Carcinoma).” She was randomized to the trial arm without trastuzumab and completed six cycles of carboplatin/paclitaxel chemotherapy. Unfortunately, a CT scan after treatment (11 October 2016) demonstrated the progression of her disease, with new pulmonary metastases, recurrent disease at vaginal apex, carcinomatosis, and worsening pelvic lymphadenopathy. As shown in Figure 1, at the time of the CT scan, the CA-125 tumor marker was 14.8 U/mL (normal range:0 to 35 U/mL), while the ctDNA copy number was 109 mutant copy/mL. She then subsequently started another clinical trial of afatinib for patients with recurrent USC overexpressing HER2. She received afatinib therapy from 13 October to 20 December 2016; unfortunately, a CT scan imaging from 6 January 2017 demonstrated progression of her disease, with increased chest adenopathy, increased size of pulmonary nodules, and vaginal cuff recurrence. At this time, the patient also described increased vaginal bleeding. A vaginal apex biopsy 12 January 2017 was consistent with vaginal recurrence with HER2 3+ expression. As shown in Figure 1, at the time of the vaginal biopsy confirming recurrent disease, CA-125 was negative at 17.2 U/mL (normal range: 0 to 35 U/mL), while ctDNA copy number was elevated to 457 mutant copy/mL. She then went on to receive trastuzumab and carboplatin for five cycles (12 January to 2 May 2017), as well as vaginal apex radiation to treat her vaginal recurrence (10 fractions, completed 11 April 2017). Unfortunately, a CT scan after her fifth cycle (19 May 2017) demonstrated interval growth of pulmonary metastases (an index lesion measured 4.4 cm, increased from 2.3 cm). With this worsening of the disease, carboplatin/trastuzumab was stopped, and she initiated treatment in an open-label clinical trial of sacituzumab govitecan for patients with recurrent uterine serous carcinoma, IMMU-132. She received two cycles of this experimental therapy but unfortunately experienced further progression of her disease (CT 10 August 2017). She was removed from the clinical trial secondary to progression, with a plan to start weekly abraxane. Unfortunately, in the 2 weeks after stopping IMMU-132, the patient developed difficulty walking and was found to have a large posterior fossa metastasis in the brain. The patient was discharged to hospice care on 31 August 2017. Notably, as described in Figure 1, the patient’s ctDNA levels were found to strongly correlate to the clinical evidence of recurrent disease as evaluated by both imaging and confirmed with biopsies, while CA-125 levels remained within normal limits and without any significant increase throughout the course of her disease.

**Figure 1 ijms-24-08873-f001:**
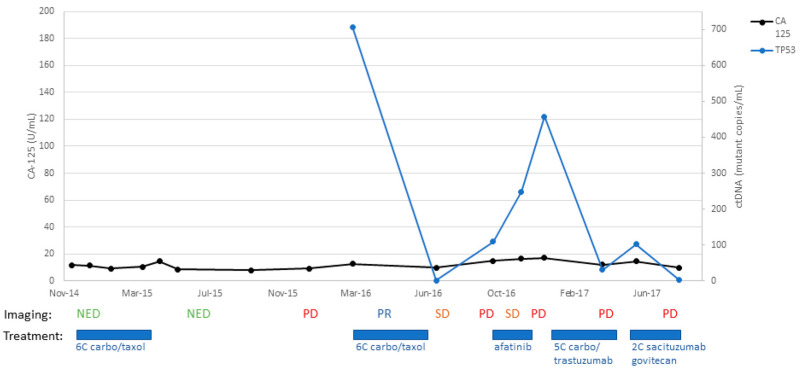
Timeline of patient #1 disease course with treatment: C/T, carboplatin/paclitaxel; PD, progression of disease; MR, mixed response; SD, stable disease; PR, partial response; NED, no evidence of disease.

Patient #3:

A 68-year-old female with stage IVB uterine serous carcinoma underwent surgical staging with total abdominal hysterectomy, bilateral salpingo-oophorectomy, subtotal omentectomy, and pelvic and periaortic lymph node biopsy on 31 March 2014. Her tumor was found to overexpress HER2, and thus she received adjuvant carboplatin, paclitaxel, and trastuzumab for six cycles. A follow-up CT scan (8 October 2015) after adjuvant therapy demonstrated no evidence of disease. She completed adjuvant brachytherapy to vaginal apex in March 2015 and completed 12 months of trastuzumab maintenance with arimidex (i.e., an aromatase inhibitor). Unfortunately, a CT scan performed on 8 October 2015 demonstrated an enlarged periaortic lymph node suspicious for recurrent disease. To treat this recurrence, she completed radiotherapy (4320cGy) on 10 February 2016; her CA-125 decreased to 23.5 U/mL (2 March 2016) from 104 U/mL (31 December 2015). Unfortunately, a subsequent CT scan (13 April 2016) demonstrated several new areas of metastatic disease in the liver and lung. She was then started on dose-dense paclitaxel with carboplatin; after three cycles, a CT scan (17 June 2016) demonstrated a partial response, with interval improvement in pulmonary metastases and hepatic implant. Imaging at the conclusion of six cycles (22 August 2016) demonstrated an overall mixed response, with decreased size of pulmonary and hepatic metastases but increased size of a mediastinal lymph node. A further mixed response was demonstrated on a subsequent CT scan (7 November 2016). Unfortunately, at this time, two brain metastases were discovered in CT and MRI imaging studies. She went on to complete gamma knife radiation to her brain lesions (22 November 2016), which remained stable-to-improved on imaging 3 months later (6 March 2017). At that time, as depicted in Figure 2, CA-125 was still within the normal limit at 34.4 U/mL (normal range: 0 to 35 U/mL), while ctDNA copy number was elevated to 46 mutant copy/mL. She completed four subsequent cycles of dose-dense paclitaxel/carboplatin. Unfortunately, a CT scan after these four cycles (6 March 2017) showed disease progression with enlarging lung metastases and increased retroperitoneal lymphadenopathy. At this point, the patient’s CA-125 was elevated to 83 U/mL while her ctDNA copy number was further increased to 69 mutant copy/mL. She then enrolled in the IMMU-132 (sacituzumab govitecan) open-label clinical trial and completed nine cycles of treatment. Imaging studies initially demonstrated a mixed response (22 June and 21 August 2017) but went on to demonstrate progressive disease (15 December 2018). She was removed from the IMMU-132 trial and was started on ixabepilone with bevacizumab. The patient required multiple delays in treatment because of leukopenia, symptomatic anemia, and dehydration. She completed three cycles of treatment (27 February 2018) and then chose to discontinue treatment due to her symptoms. The patient was discharged to hospice care (31 March 2018). As demonstrated in Figure 2, her CA-125 levels as well as ctDNA levels were found to be concordant with disease response and progression.

**Figure 2 ijms-24-08873-f002:**
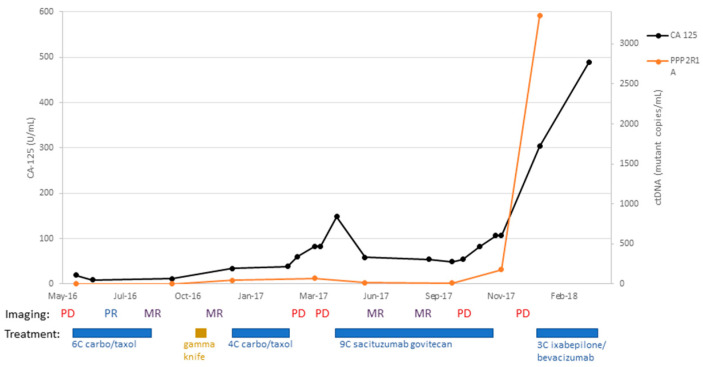
Timeline of patient #3 disease course with treatment: C/T, carboplatin/paclitaxel; PD, progression of disease; MR, mixed response; SD, stable disease; PR, partial response; NED, no evidence of disease.

Patient #9:

A 69-year-old patient underwent staging surgery for stage IVB uterine serous carcinoma (with a 20% clear cell component) on 8 August 2012. For adjuvant therapy, she received six cycles of carboplatin and paclitaxel. She completed adjuvant therapy and remained disease-free for almost 4 years. Starting in June of 2016, the patient’s CA-125 was noted to rise above its normal value (54.6 U/mL 7 June 2016). However, a CT scan acquired on 10 June 2016 did not demonstrate any visible recurrent disease. Importantly, as described in Figure 3, evaluation of ctDNA at this time point demonstrated a copy number of 114 for PIK3CA vs. a copy number of 2 for TP53 mutant copy/mL. These findings triggered a PET CT within a few weeks that demonstrated two enlarging hypermetabolic mesenteric lymph nodes, consistent with recurrent metastatic disease. She then underwent rectosigmoid resection (1 September 2016), and the final pathology demonstrated recurrent USC, HER2 2+, metastatic to the bowel. She then had several scans demonstrating no evidence of disease until an 28 November 2017 PET CT demonstrated increased avidity in a 1 cm soft tissue nodule adjacent to the ascending colon. She then started trastuzumab therapy on 19 December 2017. Notably, her CA-125 did not rise as expected with the new metastatic implant; from that point forward in her course, CA-125 was no more a reliable marker of disease status, and accordingly, ctDNA and serial imaging studies were used to guide treatment decisions. Circulating tumor DNA with a TP53 probe was present for the first two collection time points (1 mutation/mL on 8 March 2016 and 2 mutation/mL on 7 June 2016) but then was undetectable for the remaining time points, even when PIK3CA mutations were consistently increasing in patient’s plasma (Figure 3). Her subsequent imaging demonstrated stable oligometastatic disease with the 1 cm implant. Starting on 5 April 2019, her PIK3CA ctDNA values began to rise; 14 mut/mL on 5 April 2019 to 88 mut/mL on 28 June 2019. Despite this increase, there was no evidence of disease on a 17 June 2019 CT scan. She completed 31 cycles of trastzumab on 20 September 2019. Unfortunately, CT imaging on 20 September 2019 demonstrated increased size of the prior 1 cm implant, now 3.6 cm. On 28 October 2019, she underwent laparoscopic right hemicolectomy, and pathology confirmed metastatic disease; notably, HER2 was not expressed. PIK3CA ctDNA copies decreased from 1599 mut/mL before the surgery to 22 mut/mL after the surgery. She then went on to receive pembrolizumab (17 December 2019) with lenvatinib added for cycle two (12 January 2020). Notably, during the COVID-19 pandemic, lab collections were kept to a minimum and so while the patient continued treatment, we ceased ctDNA collection after 7 January 2020. Lenvatinib was discontinued in June 2020 due to side effects, and the patient continued pembrolizumab for 12 total cycles (completed 4 August 2020). A PET CT from 24 September 2020 showed new recurrent disease with an intensely avid anterior abdominal wall nodule. She was then started on the IMMU-132 trial (sacituzumab govitecan) (16 October 2020). Her subsequent CT scans demonstrated a decreased size of the lesion (11 December 2020) and stable disease (5 February 2021). On 2 April 2021, her abdominal wall metastasis was noted to have increased in size significantly, and she was removed from the IMMU-132 trial after eight cycles (31 March 2021). As the patient again had oligometastatic disease, she underwent tumor debulking of her port site metastasis on 23 April 2021. As her postop PET CT was concerning for ongoing FDG avidity along her abdominal wall, she was started on weekly temsirolimus (25 mg) on 29 June 2021 because of her PIK3CA 1007 K hotspot mutation. Her subsequent scans have all shown no evidence of disease. She remains free of disease on temsirolimus 20 mg/weekly and was last seen in our clinic on 21 February 2023.

**Figure 3 ijms-24-08873-f003:**
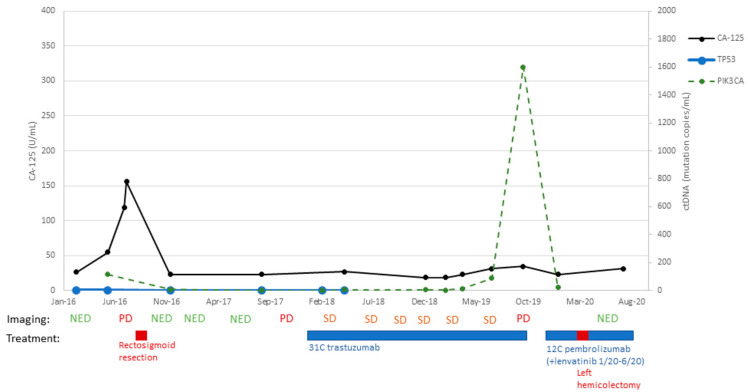
Timeline of patient #9 disease course with treatment: C/T, carboplatin/paclitaxel; PD, progression of disease; MR, mixed response; SD, stable disease; PR, partial response; NED, no evidence of disease.

Patient #13:

An 82-year-old patient underwent staging surgery on 26 September 2016 with a total laparoscopic hysterectomy, bilateral salpingo-oophorectomy, pelvic and periaortic lymph node biopsies, and omentectomy for a stage IVB uterine serous carcinoma, with HER2 overexpression. Postoperatively, she received six cycles of adjuvant carboplatin, paclitaxel, and trastuzumab. She then transitioned to maintenance trastuzumab and completed eight cycles. On 22 August 2017, she underwent a surveillance CT scan, which unfortunately demonstrated the progression of the disease. This was preceded by an increase in CA-125 to 122 (18 April 2017) from 74.7 (9 March 2017), as shown in Figure 4. She completed an additional two cycles of trastuzumab (completed 5 October 2017). On 2 November 2017, a CT scan demonstrated interval growth of lung nodules, as well as increased size of a bladder mass. She then went on to begin treatment with afatinib on 11 November 2017. She received three cycles but, unfortunately, had increasing CA-125 values and progressive disease on imaging. On 18 January 2018, she had vaginal cuff biopsies consistent with serous carcinoma. She then began treatment with the antibody-drug-conjugate trastuzumab emtansine (T-DM1) on 29 January 2018. She received 16 cycles and experienced decreased CA-125 (513 U/mL on 29 January 2018 to 37.2 U/mL on 29 March 2018). Her ctDNA was followed with two separate probes for PIK3CA and TP53, both of which decreased dramatically during her treatment with T-DM1 as well and correlated to her CA-125 trends throughout her disease course (Figure 4). She had one CT scan after initiating T-DM1 treatment on 6 July 2018, which demonstrated stable disease in the omentum and lower abdomen, with a slight increase in the size of a right upper-lobe metastasis. The patient, unfortunately, succumbed to injuries sustained in a motor vehicle accident on 21 December 2018.

**Figure 4 ijms-24-08873-f004:**
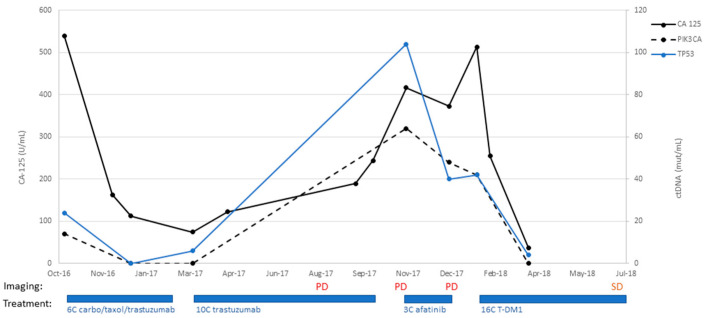
Timeline of patient #13 disease course with treatment: C/T, carboplatin/paclitaxel; PD, progression of disease; MR, mixed response; SD, stable disease; PR, partial response; NED, no evidence of disease.

Patient #14:

A 64-year-old patient underwent a staging laparoscopic hysterectomy, bilateral salpingo-oophorectomy, sentinel lymph node biopsy, and partial omentectomy on 3 January 2018. Pathology yielded stage IA carcinosarcoma of the uterus, with endometrial adenocarcinoma and homologous sarcomatous components and positive pelvic washings. She completed six cycles of adjuvant carboplatin and paclitaxel and brachytherapy to the vaginal cuff. She underwent surveillance CT scans every 3 months, as CA-125 was proven to not be a marker of disease for her. Next-generation sequencing yielded a mutation in PTEN (R130G) in her tumor. For 6 months after the completion of her adjuvant therapy, she had persistently negative results for ctDNA, CA-125, and CT imaging (Figure 5). Starting in February 2019, a 1.5 cm cystic lesion adjacent to the third portion of the duodenum was identified on a CT scan. As the patient’s ctDNA results remained negative throughout this time, and thus clinical suspicion for recurrence was low, the mass was followed with serial CT scans. The mass was unchanged on imaging until 29 April 2020 when the patient presented with partial small bowel obstruction (SBO) symptoms. On CT imaging at that time, the mass was identified to have increased in size from 1.5 cm to 1.7 cm. The patient underwent diagnostic laparoscopy and lysis of adhesions. What had appeared as a cystic mass on CT imaging, concerning for a potential recurrence, was found to be a fibrotic band causing an SBO. Multiple biopsies were sent for pathologic evaluation, all of which were negative for recurrent carcinosarcoma. This patient remains disease-free after 4 years from the initial diagnosis.

**Figure 5 ijms-24-08873-f005:**
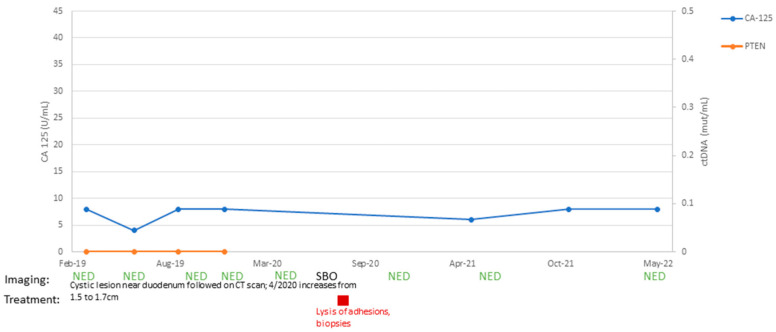
Timeline of patient #14 disease course with treatment: C/T, carboplatin/paclitaxel; PD, progression of disease; MR, mixed response; SD, stable disease; PR, partial response; NED, no evidence of disease; SBO, small bowel obstruction.

Additional Patients:

Two additional patients (i.e., patients #2 and #6) were both diagnosed with advanced CS and USC (Table 1) and were similarly treated with surgical staging/debulking followed by gold standard carboplatin and paclitaxel chemotherapy. These patients also underwent ctDNA, CA-125, and CT imaging evaluation every 6 months after completion of their adjuvant therapy and had persistently negative results for ctDNA, CA-125, and CT imaging. These two patients, for whom ctDNA was undetectable at 6 months after the completion of treatment, remain disease-free after 5 years (patient #2) and 9 years (patient #6) from the initial diagnosis.

## 3. Discussion

Growing evidence suggests that in multiple human tumors, ctDNA, a minimally invasive biomarker, can aid in the measurement of residual disease status across several settings, including after surgery and/or adjuvant treatment, during surveillance, and throughout the course of therapy for monitoring of treatment response. However, longitudinal human studies correlating ctDNA levels with tumor burden in rare, biologically aggressive uterine cancers such as USC and CS are lacking. Accordingly, in this study, we retrospectively evaluated the ability of serially collected ctDNA to screen for the clinical presence of tumor in patients with biologically aggressive USC/CS undergoing treatment and surveillance. We also compared ctDNA results with those of CT imaging (i.e., the current gold standard method for the detection of residual/recurrent disease), as well as to CA-125 serum levels, to confirm the presence or absence of tumor cells in our patient population. In this regard, while CA-125 is an FDA-approved and commonly used test for monitoring ovarian cancer patients, and several papers have demonstrated its clinical relevance for the detection of extrauterine disease in USC patients [7,8], in most patients, CA-125 remains within the normal range even while rising, complicating the interpretation of such findings [9]. Our results demonstrate the ability to sensitively detect SNV by digital PCR in the plasma of all USC/CS harboring recurrent/residual disease regardless of repetitively negative or difficult-to-interpret CA-125 testing results. These data suggest that ctDNA may confirm and, in many cases, outperform CA-125 in the monitoring of USC/CS patients. Importantly, in this regard, we found that once the patient-specific ctDNA probes were selected and validated, testing of plasma samples could be completed within hours. Thus, ctDNA evaluation could easily be incorporated into the clinical monitoring of patients with these rare uterine cancers. It is worth emphasizing, however, that while this methodology is extremely sensitive (i.e., detection limit at 0.02–0.002%) [9,10], it does require prior knowledge of tumor-specific genetic alterations in each patient’s tumor. Accordingly, in this study, we used a commercially available and FDA-approved gene panel (i.e., Foundation Medicine) [6], which targets 324 commonly mutated oncogenes and tumor suppressors to identify SNVs to generate ctDNA probes from our USC/CS patients. Of interest, during longitudinal tumor ctDNA monitoring, we discovered that one patient with recurrent USC lost genetic mutations originally identified in her primary uterine tumor, which had been considered “driver gene mutations” (i.e., TP53), while other “driver” mutations (i.e., PIK3CA) remained detectable in plasma. Importantly, CA-125 tumor shedding/detection was also lost in the recurrent tumor during disease progression. Taken together, these data suggest that clonal evolution under the selective pressure of tumor treatment may alter the genetic landscape in at least some USC/CS patients. Importantly, these findings provide support to the notion that more than one genetic probe targeting potential driver gene mutations should be selected for the longitudinal monitoring of USC and CS patients.

Molecular residual disease, which is defined as ctDNA positivity after curative surgery or chemotherapy, has been strongly associated with poor prognosis in patients with multiple tumors, including surgically resected colorectal carcinomas [11]. Our retrospective series, although limited by the small number of USC/CS patients able to be included, reports consistent results, as none of the three patients with multiple negative ctDNA tests, after the completion of their initial treatment (i.e., patients #6, #14, and #16, two out of three harboring advanced stage USC and CS), developed disease recurrence during their prolonged follow-up.

One of the goals of ultrasensitive NGS monitoring based on ctDNA is to detect cancer recurrence earlier, enabling patients to potentially benefit from surgical removal of small recurrent tumors and/or be rapidly initiated on personalized treatment modalities (i.e., customized therapeutic approaches) based on the specific driver gene mutations identified in the patient’s plasma. Moreover, in at least a subset of USC/CS patients, the use of ctDNA-based biomarkers may potentially help reduce the number of unnecessary diagnostic and/or therapeutic interventions (including frequent CT scan imaging or additional lines of chemotherapy treatment) performed during surveillance. The presence or absence of tumor-mutated ctDNA may allow distinguishment between USC/CS patients with worse vs. improved survival and/or patients with benign vs. malignant processes arising during follow-up. Consistent with this view, in a CS patient described in our series (i.e., patient #14), repetitively negative ctDNA tests helped to correctly interpret a potentially suspicious cystic mass arising in the abdomen/pelvis by CT scans. When the patient presented with partial small bowel obstruction symptoms, she underwent laparoscopic resection, and this mass was surgically confirmed to be benign in nature and, accordingly, resolved with a minimally invasive approach instead of additional lines of chemotherapy.

In conclusion, we demonstrate that serial measurements of ctDNA may represent a useful surveillance biomarker to monitor response to treatment and for the identification of early relapses in USC and CS patients. Undetectable ctDNA levels at the time of completion of the primary cancer treatment, including surgery, followed by platinum/taxane doublet chemotherapy, may provide a novel predictor of survival differences in USC/CS patients. In some patients, the use of ctDNA monitoring may also help to differentiate benign from malignant processes arising/detected by CT scan imaging during follow-up. While the use of WES and digital PCR technologies has become more accessible in recent years, widespread uptake of ctDNA monitoring would require significant technological investments by laboratories and hospitals, as well as recognition of its necessity and utility by health insurance providers. Prospective randomized trials are needed to investigate the specificity and utility of ctDNA monitoring, as well as the optimal ctDNA-guided treatment strategy for USC/CS patients.

## 4. Methods and Materials

### 4.1. Patient Enrollment and Sample Collection

Records from 16 patients, 14 harboring USC and 2 diagnosed with uterine CS, were reviewed for participation in the study after obtaining Institutional Review Board (IRB) approval. Patients’ characteristics are described in Table 1. Tumor samples were collected at the time of surgery and/or recurrence. Blood samples were collected at the time of surgery and/or throughout the patient’s clinical course. For some of the patients, surgical tumor tissue was immediately flash frozen in liquid nitrogen after harvesting, and plasma was collected in BD Vacutainer K2 EDTA (K2E) Plus Blood collection tubes (BD Diagnostics, Franklin Lakes, NJ, USA). Within four hours after collection, blood samples were centrifuged for 10 min at 1200 rpm. Plasma was then transferred to a 15 mL Falcon tube and centrifuged at 2000 rpm for an additional 10 min to eliminate any remaining cellular debris. The clinical data extracted from patient medical records included demographic variables, initial tumor site, histology, grade, stage, CA-125 levels, and chemotherapeutic regimens. Data regarding surgical procedures, sites of metastatic disease, and tumor recurrences were also collected. For several patients, CA-125 levels, in addition to computed tomography (CT) results, surgical outcomes, and chemotherapy regimens, were also available.

### 4.2. DNA Extraction Protocols

Genomic tumor DNA and germline DNA from normal matching samples were extracted from approximately 20–30 mg of tissue using the AllPrep DNA/RNA/Protein mini Kit (Qiagen GmbH, Hilden, MA, USA) according to the manufacturer’s instructions. Quantification of genomic DNA was performed using Nanodrop Spectrometer (Thermo Fisher Scientific, Waltham, MA, USA). Circulating free DNA (cfDNA) was extracted from 1 mL of plasma using the QIAamp MinElute Virus Vacuum Kit (Qiagen GmbH, Hilden) and eluted with 25 uL of DNase- and RNAse-free water.

### 4.3. Identification of Somatic Mutations

Briefly, personalized tumor mutation profiles were identified for each patient’s tumor using a clinical-grade targeted NGS assay (i.e., Foundation Medicine) [6].

### 4.4. Digital Droplet PCR (ddPCR) Analysis

Samples were prepared for ddPCR (Rain Drop Digital PCR) using TaqMan Genotyping Master Mix (Applied Biosystems, Waltham, MA, USA) and custom or inventoried TaqMan SNP Genotyping Assay (Applied Biosystems, Waltham, MA, USA), which were purchased to identify specific mutated genes including but not limited to TP53 and PIK3CA. The genotyping assays contained unlabeled PCR primers and VIC or FAM-labeled probes that specifically recognized the wild-type and mutant variants, respectively. Assays were first validated by quantitative real-time PCR (qPCR) using TaqMan Genotyping Master Mix (Applied Biosystems, Waltham, MA, USA) on a 7500 Real-Time PCR System (Applied Biosystems, Foster City, CA, USA). Specificity of the assays was confirmed by testing the tumor and matched PBMC genomic DNA of each patient. A total of 25 μL of eluted cfDNA was used for each PCR reaction, representing 1000 μL of plasma. The total PCR reaction volume was 50 μL (generating 10^6^ droplets). RainDrop Source (RainDance Technologies, Billerica, MA, USA) droplet generator and Bioer GeneTouch Thermal Cycler (Bulldog Bio, Portsmouth, NH, USA) were used for DNA amplification with the following PCR protocol: 95 °C for 10 min followed by 45 cycles of 95 °C for 15 s and 62 °C for 1 min. Droplets were read in the RainDrop Sense droplet reader (RainDance Technologies, Billerica, MA, USA) and analyzed using the RainDrop Analyst II software (RainDance Technologies, Billerica, MA, USA). The sensitivity cutoff for the ctDNA detection assay had a lower limit of 1 mutant allele in more than 1,000,000.

### 4.5. Statistical Analysis

Time-series visualization was used to assess the ability of CA-125 and ctDNA to predict the presence of tumor on CT imaging and/or at time of biopsy/surgery. No statistical inference was performed.

## Figures and Tables

**Table 1 ijms-24-08873-t001:** Characteristics of study subjects and tumor single nucleotide variants (SNVs).

Patient #	Age	Race	Stage	Grade	Histology	# ctDNA Collections	F1 SNV Mutations	ctDNA Probe
1	69	W	IB	G3	USC	8	PIK3R1 splice site 1746-2A>G, TP53 R248W, ZNF217 E451K	TP53 R248W
2	67	B	VIB	G3	USC	12	TP53 E56*	TP53 E56
3	68	W	IVB	G3	USC	8	PIK3CA N345I, MLL2 G1758fs*24, PPP2R1A P179R, TP53 S127P	PPP2R1A P179R
4	56	B	IIIB	G3	USC	3	NOTCH2 P2419fs*4, PPP2R1A P179R, TP53 C176S	PPP2R1A P179R
5	60	W	IB	G3	USC	11	TP53 R175H, PIK3R1 splice site 1746-15_1779del49	TP53 R175H
6	68	B	IIIC2	G3	USC	11	PIK3CA G106V, TP53 R248Q	TP53 R248Q
7	59	W	IA	G3	USC	5	FBXW7 R479G, PIK3CA V344G, TP53 Y220C	TP53 Y220C
8	79	W	IVB	G3	USC	4	PPP2R1A P179R, TP53 G245V	PPP2R1A P179R
9	69	W	IVB	G3	USC	11	PIK3CA G1007R, DNMT3A W860R, TAF1 R843Q, TP53 splice site 96+2T>G, TP53 V25F	TP53 V25F PIK3CA 61007R
10	67	W	IIIB	G3	USC	5	FBXW7 R441W, FBXW7 R441Q, TP53 R248Q	TP53 R248Q
11	68	B	IA	G3	USC	2	TP53 R175H, TP53 R248Q, TP53 W146*	TP53 R248Q
12	70	W	IIIC	G3	USC	8	ARID1A splice site 3406+2T>G, TP53 V157F	TP53 V25F
13	82	W	IVB	G3	USC	9	PIK3CA H1047R, MSH6 R761fs*2, TP53 R175H	TP53 R175H PIK3CA H1047R
14	65	W	IA	G3	CS	4	CTNNB1 S45F, MED12 D23Y, PTEN R130G, RB1 splice site 1049+1G>T	PTEN R130G
15	71	W	IIIC1	G3	USC	2	ARID1A R2158*, KRAS G12V, PPP2R1A P179R, TP53 M237I	PPP2R1A P179R
16	71	W	X	G3	CS	3	FBXW7 R505C, PIK3CA C407W, PAX5 A380T, TP53 splice site 920-1G>A	PIK3CA C407W

## Data Availability

All data used in this study are presented in the manuscript.

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
