# Peer review of "Monitoring Treatment Response, Early Recurrence, and Survival in Uterine Serous Carcinoma and Carcinosarcoma Patients Using Personalized Circulating Tumor DNA Biomarkers"

_ijms, 2023, doi:10.3390/ijms24108873_

Round 1
Reviewer 1 Report
This is a really interesting paper written by a leading group in this area.
The commentary (Discussion is good). In the conclusion the authors might comment on two things:
1) The potential cost of using this technology
2) The authors need to emphasize the current limitations in the final paragraph. There were several genes that were difficult to detect in the CT
Author Response
Comment 1) The potential cost of using this technology.
Author response 1) Thank you for bringing up this important point. We believe that affordability of this technology will ultimately be critical to it’s widespread uptake as a method of cancer recurrence detection. As its use in cancer recurrence is still experimental in nature, we are not sure that cost is highly relevant at this time. As evidence mounts on its reliability and specificity, and there is more widespread interest in the technology, we would expect its cost to decrease significantly. We have added language to this effect in the discussion section. (Lines 435-437)
Comment 2) The authors need to emphasize the current limitations in the final paragraph. There were several genes that were difficult to detect in the CT.
Author response 2) Thank you for bringing this to our attention. We have added further discussion of the limitations of this novel technology in lines 435-438.
Reviewer 2 Report
Overall, this is a well-written manuscript. The author showed a very detailed background and introduction. The method session was nicely presented and easy to follow. The graphics were easy to read. The discussion part was also neatly presented. I don't have any concerns to publish this article.
I don't have any questions about the English in this article.
Author Response
Comment: Overall, this is a well-written manuscript. The author showed a very detailed background and introduction. The method session was nicely presented and easy to follow. The graphics were easy to read. The discussion part was also neatly presented. I don't have any concerns to publish this article.
Author Response: Thank you.
Reviewer 3 Report
Developing new early detection strategies can greatly benefit the therapy of human diseases, especially cancers. In this study, the authors explored the use of personalized circulating tumor DNA (ctDNA) markers for monitoring the treatment response, early recurrence and survival of uterine serous carcinoma (USC) and carcinosarcoma. There are several questions/issues as below:
1) The data was collected from only 16 patients, I wonder whether the sample scale is enough to draw a conclusive result. Perhaps more data need to be collected to support the hypothesis in the study.
2) As the circulating DNA was used for analysis, the markers identified in this study may also be useful for other types of cancers. Could you share the rationale why only USC would be specifically related with such markers?
3) Whether the status of ctDNA has some correlation with the DNA in cancer cells?
Minor editing of English language required
Author Response
Developing new early detection strategies can greatly benefit the therapy of human diseases, especially cancers. In this study, the authors explored the use of personalized circulating tumor DNA (ctDNA) markers for monitoring the treatment response, early recurrence and survival of uterine serous carcinoma (USC) and carcinosarcoma. There are several questions/issues as below:
Comment 1) The data was collected from only 16 patients, I wonder whether the sample scale is enough to draw a conclusive result. Perhaps more data need to be collected to support the hypothesis in the study.
Author Response 1) Thank you for pointing out an area in which we can better clarify our conclusions. As this is a case series and not a randomized control trial, or a trial of any kind, we agree that strong conclusions cannot be drawn about the accuracy/utility of ctDNA monitoring in uterine serous carcinoma/carcinosarcoma patients. However, we do feel that the data presented in our case series is powerful and speaks to the need and interest in further studies in the future. Our hope is that with additional rigorous studies in the future, conclusive results may be drawn about this new technology in this patient population. We have edited the last sentence of the discussion accordingly. (Line 439)
Comment 2) As the circulating DNA was used for analysis, the markers identified in this study may also be useful for other types of cancers. Could you share the rationale why only USC would be specifically related with such markers?
Author Response 2) We thank the reviewer for this comment. There is no rationale for why USC alone could benefit from ctDNA monitoring. We do believe, however, that UCS and CS, given their high rates of recurrence and lack of adequate/validated serum biomarker to date, stand to greatly benefit from a technology such as this. Given that our group publishes frequently on these relatively rare gynecologic malignancies, we felt focusing on them here in this case series would be of utility and interest to the medical community. We have further clarified this viewpoint in the introduction, lines 69-71.
Comment 3) Whether the status of ctDNA has some correlation with the DNA in cancer cells?
Author Response 3) The authors have some confusion about this question and how best to answer it. There is of course a correlation between ctDNA and DNA in tumor cells; the DNA found circulating in patient serum has shed from their tumor. There is early evidence that the amount of ctDNA corresponds to stage/grade/prognosis of tumors for this reason. If there is further clarification required about this point in the paper, we would be more than happy to oblige.
Round 2
Reviewer 3 Report
The revised version of the manuscript was greatly improved. All my question were addressed.
Quality of English is nice.